# Calcium Channel Blockers Are Associated with Nocturia in Men Aged 40 Years or Older

**DOI:** 10.3390/jcm10081603

**Published:** 2021-04-09

**Authors:** Satoshi Washino, Yusuke Ugata, Kimitoshi Saito, Tomoaki Miyagawa

**Affiliations:** 1Department of Urology, Jichi Medical University Saitama Medical Center, Saitama 330-8503, Japan; lespaul991200@gmail.com (K.S.); sh2-miya@jichi.ac.jp (T.M.); 2Department of Cardiology, Jichi Medical University Saitama Medical Center, Saitama 330-8503, Japan; uga-suke@omiya.jichi.ac.jp

**Keywords:** nocturia, calcium channel blocker, anti-hypertensive agents, hypertension, blood pressure

## Abstract

Background: The associations of nocturia with hypertension and anti-hypertensive agents (AHTs) remain to be validated. Methods: This cross-sectional study examined whether blood pressure and/or frequently used classes of AHTs had consistent associations with nocturia. Methods: A total of 418 male patients aged ≥ 40 years were retrospectively assessed in terms of the International Prostate Symptom Score (IPSS), prescription medications, and blood pressure. Nocturia was evaluated using item 7 of the IPSS, and two or more episodes of nocturia per night was considered to indicate clinically important nocturia. Results: Patients taking calcium channel blockers (CCBs), but not other AHTs, experienced more episodes of nocturia than patients not taking AHTs (1.77 ± 1.07, 1.90 ± 1.19, and 1.48 ± 0.98 in CCBs alone, CCBs + other AHTs, and other AHTs alone, vs. 1.35 ± 1.08 in not taking AHTs; *p* = 0.014, *p* < 0.0001, and *p* = 0.91, respectively), whereas there was no significant difference in the number of nocturia episodes between patients with elevated and normal blood pressure. In multivariate analysis, CCB (odds ratio (OR) = 2.68, *p* < 0.0001) and age (OR = 1.06, *p* < 0.0001) were independently associated with clinically important nocturia. Conclusion: CCB was associated with nocturia, while AHTs other than CCBs and elevated blood pressure were not.

## 1. Introduction

The International Continence Society defines nocturia as the number of times an individual passes urine during their main sleep period, from the time they have fallen asleep up to the intention to rise from that period [1,2]. Nocturia affects men and women equally, but its prevalence increases with age [3,4,5]. Nocturia is associated with falls and fall-related injuries (primarily in the elderly but also in younger age groups) and reduced quality of life (mainly due to fragmented sleep and an increased prevalence of depressive symptoms, particularly in younger men and women) [5,6,7,8]. Although very common, nocturia remains underreported, undertreated, and poorly managed in adults [9].

The cause of nocturia is often multifactorial; however, distinct pathophysiological mechanisms have been delineated, including global polyuria, nocturnal polyuria, diminished bladder capacity, and primary or secondary sleep disorders. The etiology of nocturia frequently does not occur in isolation, and patients often present with many medical comorbidities, complicating both diagnosis and therapeutic management [10]. Medical therapy plays a role in the treatment of nocturia secondary to decreased bladder capacity, as lower urinary tract symptoms and overactive bladder often coincide. However, treatments focusing on increasing bladder capacity and/or reducing bladder outlet obstruction are associated with only minor clinical improvements in nocturia [10,11].

Epidemiological studies demonstrated a significant relationship between hypertension and nocturia [12,13,14,15,16]. Several studies also demonstrated that anti-hypertensive agents (AHTs), particularly calcium channel blockers (CCBs), were associated with lower urinary tract symptoms (LUTS), including nocturia [17,18,19,20,21,22]. CCB was shown to be associated with more episodes of nocturia compared to aldosterone receptor antagonists and β-blockers [23,24]. Thus, the evidence suggests associations of nocturia with hypertension and AHTs. However, these associations remain to be validated, including whether CCBs are the only type of AHT associated with nocturia.

This cross-sectional study of men aged 40 years or older examined whether blood pressure (BP) and/or frequently used classes of AHTs had consistent associations with nocturia.

## 2. Materials and Methods

### 2.1. Study Design and Patients

This retrospective cross-sectional observational study was approved by the institutional review board at Jichi Medical University Saitama Medical Center (Code RinS20-704, Date 7 September 2020). Data from male patients aged 40 years or older, who were hospitalized in the Department of Urology, Jichi Medical University Saitama Medical Center from January 2018 to December 2019 and completed the International Prostate Symptom Score (IPSS) questionnaire (Appendix A), were collected. Patients with diseases potentially affecting urinary symptoms, including T3b and T4 prostate cancer, bladder carcinoma in situ (CIS), and muscle-invasive bladder tumor, as well as those undergoing hemodialysis and those with a history of prostate surgery within 6 months before study entry or symptomatic cystitis, were excluded. A total of 487 patients were eligible, and 69 patients were excluded for the following reasons: T3b (or higher) prostate cancer T3b (*n* = 35), bladder CIS or muscle-invasive bladder tumor (*n* = 19), hemodialysis (*n* = 9), history of prostate surgery within 6 months before study entry (*n* = 5), and symptomatic cystitis (*n* = 1). A total of 418 cases were analyzed.

### 2.2. IPSS and Nocturia

All patients in this study completed the IPSS questionnaire on referral to the Department of Urology. Clinically important nocturia was defined as two or more episodes per night, according to item 7 of the IPSS. The overactive bladder symptom score (OABSS) questionnaire (Appendix A) was also collected to assess storage symptoms.

### 2.3. Medications

Use of prescription medications was captured by self-reporting and direct observation of medication labels by pharmacists in our institute. We recorded use of the six classes of AHTs most commonly prescribed in our population: CCBs (dihydropyridine type), angiotensin receptor blockers (ARBs), angiotensin-converting enzyme inhibitors (ACE-Is), β-blockers, α-blockers, and thiazide diuretics.

### 2.4. Measurement of BP and Variation between Daytime and Morning BP

Elevated BP was defined as systolic blood pressure (SBP) >140 mmHg or diastolic blood pressure (DBP) >90 mmHg in the daytime. Daytime BP was measured between 10:00 and 14:00 on the first day of hospitalization (the day before surgery), and morning BP was measured between 06:00 and 07:00 the following day (the day of surgery).

### 2.5. Covariates

Diabetes mellitus (DM) was defined as glycated hemoglobin >6.2% or taking hypoglycemic agents. The sum of the scores on IPSS items 1–6 was used to define LUTS other than nocturia. Other covariates included age, body mass index (BMI), and the estimated glomerular filtration ratio (eGFR). Glycated hemoglobin and eGFR were assessed within 4 months before admission.

### 2.6. Endpoints

The primary endpoint was whether AHTs and/or elevated BP were associated with nocturia. The secondary endpoint was which types of AHTs were associated with nocturia.

### 2.7. Statistical Analysis

All data are shown as the mean and ± standard deviation (SD) unless otherwise indicated. Variables were compared using the Mann–Whitney U test and Fisher’s exact test, as appropriate. Comparisons among three or more groups were performed using the Kruskal–Wallis test with Dunn’s multiple comparison test. Spearman’s rank correlation coefficient was used to calculate correlations. Binary logistic regression analysis was performed to identify factors independently associated with clinically important nocturia. Statistical analyses were performed using GraphPad Prism (ver. 7.0; GraphPad, La Jolla, CA, USA) and SPSS Statistics for Windows software (ver. 19.0; IBM Corp., Armonk, NY, USA).

In all analyses, *p* < 0.05 was taken to indicate statistical significance.

## 3. Results

### 3.1. Patient Characteristics

The mean age of the study population was 69.4 ± 7.6 years. The reasons for admission were as follows: prostatectomy for prostate cancer (*n* = 250, 60%), prostate biopsy for suspected prostate cancer (*n* = 86, 21%), partial or radical nephrectomy for kidney cancer (*n* = 30, 7%), nephroureterectomy for renal pelvic/ureteral cancer (*n* = 17, 4%), and others (*n* = 12, 3%) (Table 1).

The mean duration from reporting IPSS questionnaire to assessing medications and BP was 2.71 ± 1.60 months. A total of 240 patients (57%) were taking AHTs, including CCBs (*n* = 170, 71%), ARBs (*n* = 141, 59%), ACE-Is (*n* = 30, 13%), β-blockers (*n* = 30, 13%), α-blockers (*n* = 30, 13%), and thiazide diuretics (*n* = 20, 8%). The 107, 94, and 39 patients were taking single, dual, and more than three types of AHTs, respectively. Daytime BP was elevated in 113 patients (27%). A total of 100 patients (24%) were classified as having DM, 61 of whom were taking hypoglycemic agents.

The mean total IPSS and mean score on item 7 (nocturia) were 8.23 ± 6.88 and 1.58 ± 1.11, respectively. The mean score on item 7 increased with age (1.28 ± 1.12, 1.54 ± 1.09, and 2.08 ± 1.04 in the 40–65, 66–75, and ≥76 years age groups, respectively) (Table 2).

### 3.2. Association of Nocturia with BP and AHTs

There was no significant difference in nocturia episode frequency between patients with elevated daytime BP and normotensive individuals (1.46 ± 0.98 and 1.63 ± 1.16 respectively, *p* = 0.14, (Figure 1A), regardless of AHT status (Figure 1B).

Patients taking AHTs experienced significantly more episodes of nocturia compared to those not taking AHTs (1.75 ± 1.12 and 1.35 ± 1.08 respectively, *p* = 0.0003, Figure 2A). Patients taking CCB alone or CCB plus other AHTs experienced more episodes of nocturia (1.77 ± 1.07 or 1.90 ± 1.19, respectively) compared to patients not taking AHTs (1.35 ± 1.08, *p* = 0.014 and *p* < 0.0001, respectively), while the difference in number of nocturia episodes did not differ between patients taking AHTs other than CCB and those not taking AHTs (1.48 ± 0.98 and 1.35 ± 1.08 respectively, *p* = 0.91, Figure 2B). The effect of CCB on nocturia was prominent in younger patients (40–65 years) (0.96 ± 0.88 in patients not taking CCB vs. 2.00 ± 1.27 in those taking CCB respectively, *p* < 0.0001), although a difference was still evident in older patients ≥ 76 years) (1.84 ± 0.95 vs. 2.29 ± 1.08 respectively, *p* = 0.036) (Figure 2C). The time of CCB taking (daytime, evening, or both) did not significantly affect the number of nocturia episodes (1.79 ± 1.16, 2.12 ± 0.78, or 2.14 ± 1.28, *p* = 0.27, Figure 2D).

### 3.3. Association of Lower Urinary Tract Symptoms with AHTs

In IPSS, the scores for urgency and frequency were higher in patients taking CCB plus other AHTs compared to those not taking AHTs (*p* = 0.02 and *p* = 0.06, respectively) (Figure 3). The scores for weak stream and straining were more in patients taking AHTs other than CCB compared to in those not taking AHTs (*p* = 0.05 and *p* = 0.005). The OABSS questionnaire was available in 388 patients (93%). The scores for nighttime frequency, urgency, and urgency incontinence were higher in patients taking CCB alone or CCB plus other AHTs compared to in those not taking AHT in OABSS (*p* = 0.004 or *p* = 0.0007 in nighttime frequency, *p* = 0.02 or *p* = 0.02 in urgency, *p* = 0.09 or *p* = 0.09 in urgency incontinence, respectively) (Figure 4).

### 3.4. Correlations between Nocturia and Other Covariates

There were positive correlations between the number of nocturia episodes and age (*r* = 0.27, *p* < 0.0001) and the sum of the scores on IPSS items 1–6 (*r* = 0.25, *p* < 0.0001), and a very weak correlation between the number of nocturia episodes and the glycated hemoglobin level (*r* = 0.13, *p* = 0.006). In contrast, no correlation was detected between nocturia and BMI (*p* = 0.60) or eGFR (*p* = 0.15) (Appendix A). The presence of LUTS (the sum of the scores on IPSS items 1−6 more than 6) was associated with nocturia regardless of age group (Appendix A).

### 3.5. Factors Associated with Clinically Important Nocturia in Multivariate Analysis

A total of 190 patients exhibited clinically important nocturia, while 227 patients did not. In univariate analysis, age, eGFR, the sum of the scores on IPSS items 1–6, CCBs, and β-blockers were associated with clinically important nocturia, while BMI, elevated daytime and morning BP, glycated hemoglobin level, ARBs, ACE-Is, α-blockers, and thiazide diuretics were not (Table 3). In multivariate analysis, CCBs (odds ratio (OR) = 2.68, *p* < 0.0001), age (OR = 1.06, *p* < 0.0001), and the sum score for IPSS items 1–6 (OR = 1.05, *p* < 0.0001) were independently associated with clinically important nocturia.

## 4. Discussion

In the present study, we assessed the association of nocturia with frequently used classes of AHTs and other covariates, including BP, in more than 400 patients, demonstrating that CCB use, but not use of AHTs other than CCB nor BP elevation itself, was associated with nocturia.

Although nocturia has often been treated with agents targeting benign prostatic hyperplasia and/or overactive bladder, such treatment has been shown to be beneficial only in certain patients [10]; recently, other factors were considered to be more closely related to the development of nocturia [11]. Nocturia is closely associated with nocturnal polyuria, the causes of which vary widely and include hypertension, chronic heart failure, chronic kidney disease, peripheral edema, diabetes insipidus, and uncontrolled DM [25].

Recently, associations of hypertension and AHTs with nocturia were demonstrated [15,16,17,18,19,20,21,22,23,24]. Especially, CCB use was reported to be positively associated with nocturia [18,19,20,21,22,23,24]. Bulpit et al. and Hollenberg et al. demonstrated that the CCBs amlodipine and/or nifedipine promote nocturia and increased urination compared to the aldosterone antagonist eplerenone or β-blocker bisoprolol [23,24], while Hall demonstrated that CCB use as monotherapy or combination with other AHTs, but not AHTs other than CCBs, was significantly associated with increased prevalence of nocturia (OR 2.65) in women aged less than 55 but not in those aged 55 or more [19]. Elhebir et al. demonstrated that CCBs were more likely to suffer from severe LUTS (OR 12.45 in Male and 7.75 in Female), including nocturia [18]. Another study demonstrated a significant increase in the mean IPSS score in CCB-users as compared to untreated hypertensives [26]. Interestingly, this study reported that angiotensin receptor blockers may have the potential to improve LUTS in men. The present study supported the association of nocturia with CCBs. We assessed the association with nocturia not only of CCB use, but also of use of AHTs other than CCBs, and other covariates which are considered to be associated with nocturia/nocturnal polyuria [25], demonstrating that CCBs were the only AHTs independently associated with clinically important nocturia (Table 3), while AHTs other than CCBs were not likely to be significantly associated with nocturia (Figure 2B, multivariate analysis in Table 3). These results suggest that the anti-hypertensive effect itself does not affect nocturia, and there are likely to be specific mechanisms through which CCBs promote nocturia. The CCBs including amlodipine and nifedipine promote not only nocturia, but also peripheral edema [23,24,27]. Peripheral edema is associated with nocturnal polyuria and nocturia [25,28], and therefore CCBs-associated peripheral edema may lead to nocturnal polyuria and nocturia. Santiapillai et al. reported that withdrawal of CCB improved the nocturia as well as peripheral edema in 7 of 22 patients who experienced nocturia and were taking CCBs [22]. Interestingly, the existence of CCB-associated lower limb edema was not predictive of the nocturia response to CCB withdrawal: it was present in four out of seven patients whose nocturia significantly improved on CCB withdrawal and in four out of eight whose nocturia did not improve; however, the edema resolved in all cases. On the other hand, Elhebir et al. reported that CCBs affected not only nocturia, but also storage symptoms and, to a lesser extent, voiding symptoms [18]. The present study demonstrated that CCBs with or without other AHTs affected storage symptoms, such as urgency and frequency, in the present study (Figure 3 and Figure 4). CCBs could affect the bladder and/or prostate function, which may be one of the mechanisms underlying their association with nocturia. However, it remains undetermined how CCBs are associated with nocturia and further studies are required to determine the underlying mechanisms. In the present study, the effect of CCB on nocturia was prominent in younger patients compared to older patients (Figure 2C). In older patients, nocturia could have multiple antecedents, such that the association of CCBs with nocturia would be relatively weaker. Accordingly, the number of nocturia episodes increased with age in patients not taking CCBs (Figure 2C).

Yokoyama et al. demonstrated an association between nocturnal polyuria and untreated hypertension in women [15]. Meanwhile, Victor reported that uncontrolled hypertension was independently associated with clinically important nocturia in black men aged 35–49 years [16]. In contrast, BP elevation was not associated with nocturia in the present study (Figure 1A,B). These discrepancies may have been partly due to differences in the study populations.

This study had some limitations. In particular, the retrospective design may have led to selection bias. The number of patients was relatively small in some groups, especially the number of patients taking ACE-Is, β-blockers, α-blockers, and thiazide diuretics. Also, most patients had urological cancers, which may have affected the LUTS, including nocturia. BP measurements were obtained in the hospital and there were 2.7 months of lags from reporting IPSS questionnaire to assessing medications or BP, which might affect results in the present study. Finally, nocturia episodes were determined based on the IPSS questionnaire in the present study, although a frequency volume chart is more reliable for determining nocturia [29].

In conclusion, CCBs were the only AHTs associated with nocturia in this study, especially in the younger patients, while BP elevation itself showed no such association. CCBs were also associated with storage symptoms. Urologists and physicians should take the association between CCB and nocturia into consideration when starting or adding AHTs and evaluating or treating nocturia.

## Figures and Tables

**Figure 1 jcm-10-01603-f001:**
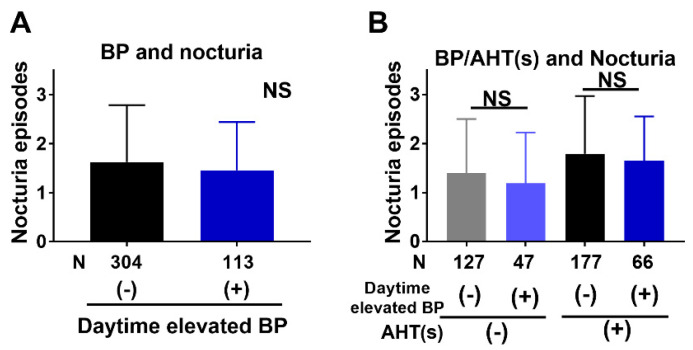
Nocturia was not associated with elevated BP. Daytime elevated BP was not associated with nocturia episodes (**A**), which was irrespective of taking AHTs or not (**B**). (**A**) Black and blue columns represent average of nocturia episodes in patients without and with BP elevation, respectively. (**B**) Gray and light blue columns represent average of nocturia episodes in patients without and with BP elevation in patients that do not take AHTs while black and dark blue columns represent those taking AHTs, respectively. BP: blood pressure, AHT(s): antihypertensive agents, NS: not significant, CCB: calcium channel blocker.

**Figure 2 jcm-10-01603-f002:**
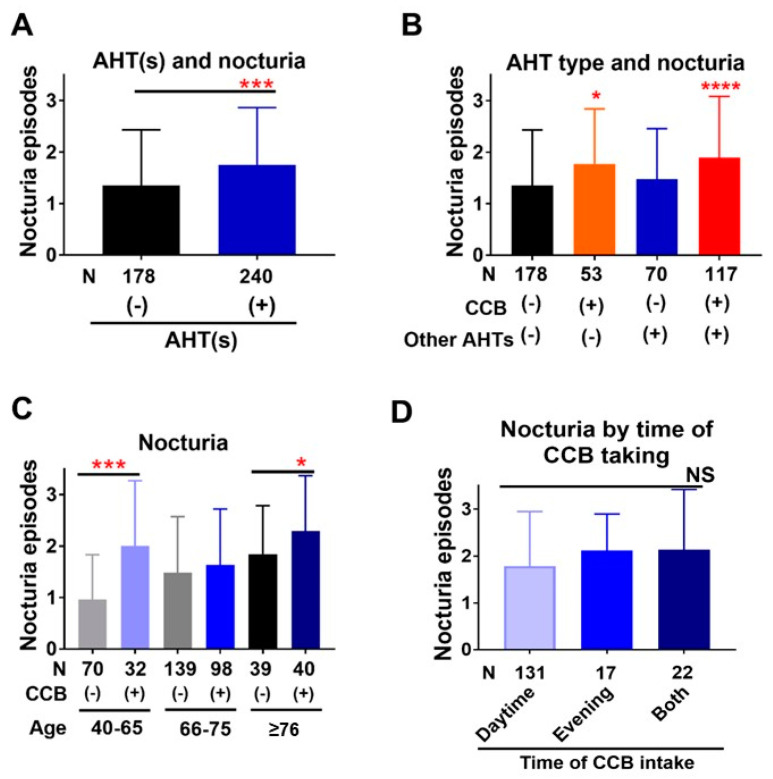
Nocturia was associated with CCBs, but no other types of AHTs. (**A**) Nocturia episodes were significantly higher in patients taking AHT(s) (blue column) compared to those not taking AHT(s) (black column). (**B**) Patients taking CCBs, but no other AHT(s), experienced more nocturia episodes compared to those not taking any AHT. Black, orange, blue, and red columns represent average of nocturia episodes in patients taking no AHTs, only CCB, AHTs other than CCB, and CCB plus other AHTs, respectively. (**C**) The association between nocturia episodes and the CCB used was significant in younger patients. Light gray, gray and black columns represent average of nocturia episodes in patients not taking CCB with age raging from 40−65, 66−75, and 76 or more respectively, while light blue, blue and dark blue columns represent the same age groups in patients taking CCB. (**D**) The time at which CCB was administered did not significantly affect the number of nocturia episodes. Light blue, blue and dark blue columns represent the average of nocturia episodes in patients taking CCBs during daytime, evening or both, respectively. AHT(s): antihypertensive agents, NS: not significant, CCB: calcium channel blocker. * *p* < 0.05, *** *p* < 0.001, and **** *p* < 0.0001.

**Figure 3 jcm-10-01603-f003:**
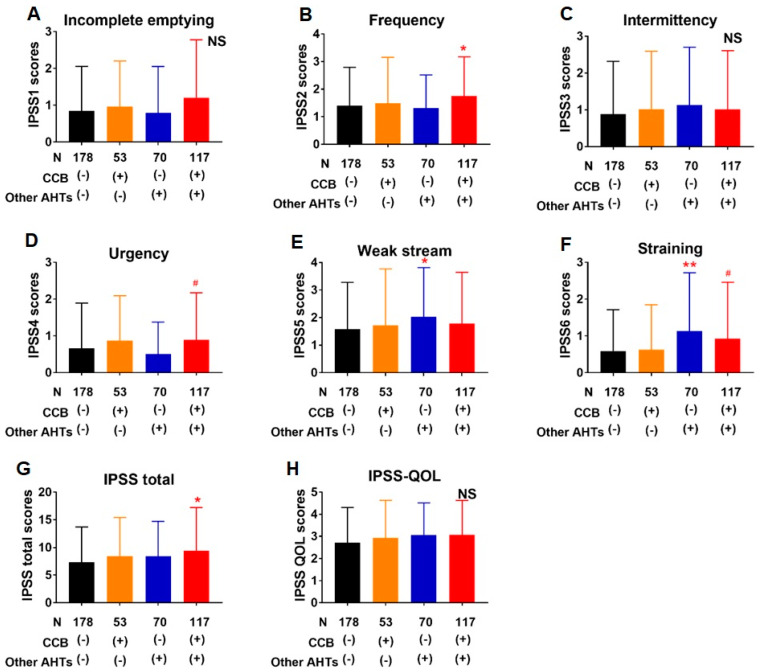
Comparison of respective IPSS items other than IPSS item 7, nocturia ((**A**) incomplete emptying, (**B**) frequency, (**C**) intermittency, (**D**) urgency, (**E**) weak stream, and (**F**) straining), IPSS total score (**G**) and IPSS-QOL (**H**) in patients taking CCB alone (*n* = 53, orange columns), other AHT(s) (*n* = 70, blue columns), and CCB plus other AHT(s) (*n* = 117, red columns) compared to those not taking AHTs (*n* = 178, black columns). Frequency (**B**), urgency (**D**), straining (**F**), and IPSS total score (**G**) were higher in patients taking CCB plus other AHTs compared to those not taking AHTs. Weak stream (**E**) and straining (**F**) were higher in patients taking AHTs other than CCB compared to those not taking AHTs. AHT(s) antihypertensive agents, CCB calcium channel blocker, IPSS international prostate symptom score, QOL quality of life, NS not significant. # *p* < 0.1, * *p* < 0.05, ** *p* < 0.01.

**Figure 4 jcm-10-01603-f004:**
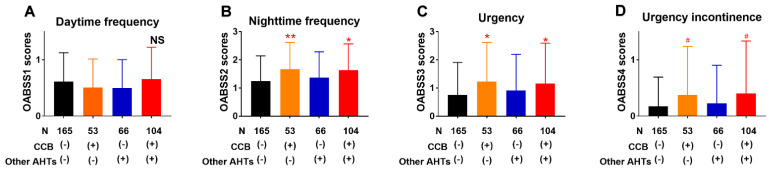
Comparison of respective OABSS items ((**A**) daytime frequency, (**B**) night time frequency, (**C**) urgency, and (**D**) urgency incontinence) in patients taking CCB alone (*n* = 53, orange columns), other AHTs (*n* = 66, blue columns), and CCB plus other AHTs (*n* = 104, red columns) compared to those not taking AHTs (*n* = 165, black columns). Nighttime frequency (**B**), urgency (**C**), and urgency incontinence (**D**) were higher in patients taking CCB alone and CCB plus other AHTs. Patients not taking AHTs other than CCB, were comparable to those not taking any AHTs. AHT(s) antihypertensive agents, CCB calcium channel blocker, OABSS overactive bladder symptom score, NS not significant, # *p* < 0.1, * *p* < 0.05, ** *p* < 0.01.

**Table 1 jcm-10-01603-t001:** Patients’ characteristics.

	*n*	%
Age, mean (SD)	69.4	(7.60)
40–65	102	(24)
66–75	237	(57)
76 or more	79	(19)
Diseases for hospitalization		
Prostate cancer	250	(60)
Prostate cancer suspected	86	(21)
Renal cell carcinoma	30	(7)
Bladder tumor	20	(5)
Renal pelvic/ureteral Ca	17	(4)
Others	12	(3)
Taking AHTs	240	(57)
Daytime elevated BP	113	(27)
DM	100	(24)
Taking hypoglycemic agent	61	(15)
BMI, kg/m^2^, mean (SD)	24.3	(3.22)
eGFR, mL/min/1.73 m^2^, mean (SD)	71.5	(16.5)
Glycated hemoglobin, %, mean (SD)	5.97	(0.695)

Abbreviations: Ca: carcinoma, AHTs: antihypertensive agents, BP: blood pressure, DM: diabetes mellitus, BMI: body mass index, eGFR: estimated glomerular filtration ratio, SD: standard deviation.

**Table 2 jcm-10-01603-t002:** IPSS according to age groups.

IPSS	40–65 Years (*n* = 102)	66–75 Years (*n* = 237)	≥76 Years (*n* = 79)	Total (*n* = 418)
IPSS total, mean ± SD	6.87 ± 5.96	8.69 ± 7.38	8.62 ± 6.27	8.23 ± 6.88
0–7	64%	57%	53%	58%
8–19	32%	34%	39%	35%
20–35	4%	10%	8%	8%
IPSS item 7 (Nocturia), mean ± SD	1.28 ± 1.12	1.54 ± 1.09	2.08 ± 1.04	1.58 ± 1.11
IPSS-QOL, mean ± SD	2.69 ± 1.57	2.83 ± 1.57	3.36 ± 1.61	2.89 ± 1.59

Abbreviations: IPSS: International Prostate Symptoms Score, SD: standard deviation, QOL: quality of life.

**Table 3 jcm-10-01603-t003:** Univariate and multivariate analysis to analyze factors associated with clinically important nocturia.

Variables		Univariate Analysis			Multivariate Analysis		
		OR	95% CI	*p* Value	OR	(95% CI)	*p* Value
Age	years	1.07	(1.04–1.10)	<0.0001	1.06	(1.03–1.10)	<0.0001
BMI	kg/m^2^	1.04	(0.90–1.19)	0.632	(-)		
Daytime elevated BP	mmHg	0.97	(0.63–1.50)	0.889	(-)		
Morning elevated BP	mmHg	0.93	(0.61–1.43)	0.932	(-)		
Glycated Hb	%	1.02	(0.80–1.29)	0.885	(-)		
eGFR	mL/min/ 1.73 m^2^	0.99	(0.98–1.00)	0.048	1.00	(0.98–1.01)	0.741
Sum of IPSS item 1–6		1.05	(1.01–1.07)	0.006	1.05	(1.02–1.09)	<0.0001
AHTs	CCB	2.80	(1.88–4.18)	<0.0001	2.68	(1.69–4.26)	<0.0001
	ARB	1.48	(0.99–2.22)	0.059	0.89	(0.54–1.46)	0.638
	ACE-I	0.72	(0.34–1.51)	0.385	1.05	(0.46–2.37)	0.916
	β-blocker	2.17	(1.00–4.68)	0.048	2.15	(0.94–4.90)	0.069
	α-blocker	1.86	(0.87–3.98)	0.107	1.59	(0.68–3.72)	0.282
	Thiazide	1.49	(0.60–3.67)	0.388	0.84	(0.31–2.26)	0.725

Abbreviations: BMI: body mass index, BP: blood pressure, Glycated Hb: glycated hemoglobin, eGFR: estimated glomerular filtration ratio, IPSS: International Prostate Symptom Score, AHTs: antihypertensive agents, OR: odds ratio, CI: confidence interval, CCB: calcium channel blocker, ARB: angiotensin receptor blocker, ACE-I: angiotensin converting enzyme inhibitor, (-) not assessed.

## Data Availability

The data are available on request by contacting the corresponding author (S.W.).

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
