# Peer review of "Calcium Channel Blockers Are Associated with Nocturia in Men Aged 40 Years or Older"

_jcm, 2021, doi:10.3390/jcm10081603_

Round 1

Reviewer 1 Report

Clearly organized with statistical presentations without confusion. two minor points: Lines 217-218 are missing two words   and    line 259  a misspelling.

The reader might be interested in a statement such as "Calcium channel blocker and other medications like hydralazine and aminophylline that dilate afferent  renal arterioles may increase urine output by increasing renal blood flow. Mineralocoticoid receptor blockers like spironolactone and eplurenone which increase levels of aldosterone (reference 23) would not increase urine output.  Angiotensin converting enzyme inhibitors and angiotensin receptor blockers employ more complex mechanisms that do not result in an increase in urine flow."

Author Response

Thank you so much for reviewing our manuscript.

We really appreciate your comments. We revised manuscript according to comments from reviewers including you. Please check out our revised manuscript.

  1. Line 217-218, missing two words

Response: Could you let me know what is missing? We were not able to find the missing words.

  1. Line 259, misspelling

Response: We revised it accordingly.

  1. The reader might be interested in a statement such as "Calcium channel blocker and other medications like hydralazine and aminophylline that dilate afferent renal arterioles may increase urine output by increasing renal blood flow. Mineralocoticoid receptor blockers like spironolactone and eplurenone which increase levels of aldosterone (reference 23) would not increase urine output.  Angiotensin converting enzyme inhibitors and angiotensin receptor blockers employ more complex mechanisms that do not result in an increase in urine flow.

Response: I agree this reviewer’s comments might be one of mechanisms how CCB use increases nocturia. However, the present study did not demonstrate an increase in urine output in patients taking CCBs (we demonstrated an increase of nocturia episodes), and therefore we are reluctant to put a statement about how different types of antihypertensive agents affect urine output.

Reviewer 2 Report

This is an important paper that highlights the importance of the holistic assessment of patients with nocturia. The data presented herein support an association of CCB intake with nocturia, in a slightly unusual patient cohort that consisted of patients with uro-oncological conditions. CCBs are one of the most common types of antihypertensive medications, which are typically prescribed as 1st/2nd line therapy, their full side-effect profile, particularly the association with lower urinary tract symptoms until recently has been unclear. The association shown by authors supports the notion that a change in anti-hypertensive therapy is a valid strategy in the management of nocturia patients.

Author Response

Thank you so much for reviewing our manuscript. Your comments encourage us to perform further studies in this field. We revised our manuscript according to other reviewer’s comments. Please check it out.

Reviewer 3 Report

The authors should be commended for their study

  1. The authors used blood pressures taken as an inpatient for this study. Inpatient blood pressures on the first day of admission after a major surgery do not accurately reflect patient's normal daytime BP. As such, I would be reluctant to draw any conclusions from theses blood pressures.
  2. A major limitation of the study is that blood pressures were measured at a different time than the completion of the IPSS questionnaires by 2.7 months.  
  3. Similar to #2, medication reconciliation was completed at a different time than the completion of the IPSS questionnaire by 2.7 months. It is unclear if and how the authors confirmed use of these medications at the time of completion of the questionnaire.
  4. A voiding diary would provide a better analysis than the IPSS  
  5.  In the abstract, the results need to be better stated. One statement states that "patient taking AHTs experienced significantly more episodes of nocturia than those not taking AHTs." Two statements later, "patients taking CCb, but not other AHTs, experienced more episodes of nocturia than patients not taking AHTs." The first statement is misleading.
  6. The authors fail to explain their findings regarding nocturia and CCB use based on age. Why was there a difference in the younger and older patient groups but not in the 66-75 year old category?
  7. The authors should clarify what their study adds to the current literature and how their study is different from the other studies that have shown an association between CCB use and nocturia

Author Response

Thank you so much for reviewing our manuscript.

We really appreciate your comments. We revised our manuscript according to comments of reviewers including you. Please check our revised manuscript.

  1. The authors used blood pressures taken as an inpatient for this study. Inpatient blood pressures on the first day of admission after a major surgery do not accurately reflect patient's normal daytime BP. As such, I would be reluctant to draw any conclusions from theses blood pressures

.

Response: I’m sorry that the statement in the time of BP measurement was not clear. BPs measurement was performed before surgeries in all patients. Our patients hospitalize one day before surgery and the following days are the day performing surgery. The first day of hospitalization means the day before surgery and the morning in the following day means the morning before surgery. We revised our manuscript accordingly. Please check sentences in Line 84-85. Inpatient BP would be somewhat different with outpatient BP and this is the limitation of this study. We put a statement about this limitation in Line 260-262.

  1. A major limitation of the study is that blood pressures were measured at a different time than the completion of the IPSS questionnaires by 2.7 months.  
  2. Similar to #2, medication reconciliation was completed at a different time than the completion of the IPSS questionnaire by 2.7 months. It is unclear if and how the authors confirmed use of these medications at the time of completion of the questionnaire.

Response: We are recognizing that these are limitations of the present study. However, LUTS including nocturia usually do not change acutely unless urinary tract infection occurs or surgeries for prostate are performed. We excluded patients who suffered symptomatic cystitis and had history of prostate surgery within 6 months before study entry. Please see Line 67-68.

The medication reconciliation was critical in the present study. At the time of IPSS completion we, medical doctors, checked the medications in some patients whereas in other patients that is missing. Therefore we used the data of medication reconciliation when patients were hospitalized, which was captured by pharmacist and was more precise and complete than that by medical doctors. We are putting a statement about this limitation in the Discussion section. Please see Line 260-262.

  1. A voiding diary would provide a better analysis than the IPSS

Response: We are recognizing this issue is one of limitations in the present study. We need to use a voiding diary to assess nocturia and nocturnal polyuria in future studies. We are putting a statement about this limitation in the Discussion section. Please see Line 262-264.

  1. In the abstract, the results need to be better stated. One statement states that "patient taking AHTs experienced significantly more episodes of nocturia than those not taking AHTs." Two statements later, "patients taking CCb, but not other AHTs, experienced more episodes of nocturia than patients not taking AHTs." The first statement is misleading.

Response: We agree with the reviewer’s comments. We revised the manuscript accordingly. Please check Line 14-19.

  1. The authors fail to explain their findings regarding nocturia and CCB use based on age. Why was there a difference in the younger and older patient groups but not in the 66-75 year old category?

Response: That is a point but it’s hard for us to explain why the difference did not exist in the age of 66-75 while that in =< 65 or >75 years old category existed. Hall SA also demonstrated that CCB use as monotherapy or combination with other AHTs was significantly associated with increased prevalence of nocturia in women aged less than 55 but not in those aged 55 or more. Younger age seems to be more affected by CCBs, which is stated in Line 245-249. It would be necessary to if and how age is associated with the CCBs effect on nocturia or nocturnal polyuria in a larger cohort in the future.

  1. The authors should clarify what their study adds to the current literature and how their study is different from the other studies that have shown an association between CCB use and nocturia

Response: Basically, the present study concrete the association of CCB with nocturia as previous studies demonstrated. The unique point of our study is that we assessed the association with nocturia not only of CCB use, but also of use of AHTs other than CCBs, and other covariates which is considered to be associated with nocturia/ nocturnal polyuria, demonstrating that CCB use was independently associated with clinically important nocturia. We also demonstrated that CCB use affect storage symptoms, which might be the cause of nocturia. The different points in this study with previous studies are BP elevation was not associated with nocturia. We reorganized the Discussion section to define the characteristics of our study. Please see Line 199-202, Line 221-226, and Line 239-241.